# Okara-Enriched Gluten-Free Bread: Nutritional, Antioxidant and Sensory Properties

**DOI:** 10.3390/molecules28104098

**Published:** 2023-05-15

**Authors:** Mirjana B. Pešić, Milica M. Pešić, Jelena Bezbradica, Anđela B. Stanojević, Petra Ivković, Danijel D. Milinčić, Mirjana Demin, Aleksandar Ž. Kostić, Biljana Dojčinović, Sladjana P. Stanojević

**Affiliations:** 1Institute of Food Technology and Biochemistry, Faculty of Agriculture, University of Belgrade, 11080 Belgrade, Serbia; mpesic1801@gmail.com (M.M.P.);; 2Lund University Center for Sustainable Studies (LUCSUS), Faculty of Social Sciences, 223 62 Lund, Sweden; 3Institute of Chemistry, Technology and Metallurgy, National Institute of the Republic of Serbia, University of Belgrade, Njegoševa 12, 11000 Belgrade, Serbia

**Keywords:** okara, sustainable food production, buckwheat, rice, millet, phenolics, antioxidant properties

## Abstract

The aim of this study was to produce an eco-innovative gluten-free bread with a pleasant taste and a unique formulation that includes the highest quality grains and pseudocereals (buckwheat; rice; and millet); and okara; a by-product of soy milk production. The mixture of pseudocereal and cereal flour contained buckwheat flour 45%, rice flour 33%, and millet flour 22%. Three gluten-free breads; each containing different contents of gluten-free flour (90%, 80%, and 70%, respectively); okara (10%, 20%, and 30%, respectively); and a control sample (without okara); were prepared and subjected to sensory evaluation. The okara-enriched gluten-free bread with the highest sensory score was selected for further analysis of physico-chemical (total proteins; total carbohydrates; insoluble fiber; soluble fiber; sugars; total lipids; saturated fatty acids; and salt) and functional properties (total phenolic content and antioxidant properties). The highest sensory scores were obtained for 30% okara-enriched gluten-free bread including taste; shape; odor; chewiness; and cross-section properties; classifying this bread in the category of very good quality and excellent quality (mean score 4.30 by trained evaluators and 4.59 by consumers). This bread was characterized by a high content of dietary fiber (14%), the absence of sugar; low content of saturated fatty acids (0.8%), rich source of proteins (8.8%) and certain minerals (e.g.,; iron; zinc); and low energy value (136.37 kcal/100g DW). Total phenolic content was 133.75 mgGAE/100g FW; whereas ferric reducing power; ABTS radical cation; and DPPH radical scavenging activity were 119.25 mgAA/100g FW; 86.80 mgTrolox/100g FW; and 49.92 mgTrolox/100g FW; respectively. Okara addition in gluten-free bread production enables the formulation of high-nutritive; good antioxidative; low-energy bread; and better soy milk waste management.

## 1. Introduction

Gluten-free bread, according to its composition and method of preparation, belongs to the group of dietary foods [1] and is primarily intended for the nutrition of people suffering from celiac disease, who often have difficulties in applying adequate nutrition [2]. Celiac disease is a lifelong dietary disorder, present worldwide, and defined as “immuno-mediated enteropathy triggered by the ingestion of gluten in susceptible patients” [3,4]. Celiac disease can appear without visible symptoms, but much more often there are symptoms such as an inflammatory disorder of the small intestine (due to difficulties in absorption of many nutrients, such as liposoluble vitamins, folic acid, and iron), abdominal discomfort, weight loss, diarrhea, osteoporosis, fatigue, disorders of multiple organ systems, and an increased risk of some cancers [2,3,4,5]. In the case of celiac disease, a gluten-free diet is recommended [2,3,4].

In the preparation of gluten-free bread, starches from gluten-free grains (corn and rice) and hydrocolloids of natural origin are most often used, as well as buckwheat and rice flour [6,7]. In a lower extent, the flour of millet can also be used [8]. Commonly used commercial buckwheat flour is a nutritious, rich source of starch (~60–73%) and protein (~7–38%) with well-balanced amino acid content (glutamic and aspartic acid, arginine, leucine, glycin, serine, phenylalanine, alanine, and proline) [9,10,11]. Buckwheat flour also contains a relatively high level of dietary fiber (~3–10%), soluble carbohydrates (~1%), lipids (~1–4%; with dominant fatty acids such as linoleic, oleic, and palmatic acid), minerals (~1–3%, such as Mg, P, K, Ca, Fe, and Zn) vitamins and pseudovitamins (B9, B3, K, and choline), and compounds such as organic acids, tannins, nucleotides, and nucleic acids [8,9,10]. Buckwheat can also be used as a good source of polyphenols (rutin, catechin, quercetin, and hyperin), and natural antioxidants that are absent in other pseudocereals or grains [11,12]. Consumption of buckwheat in a diet can have several health benefits such as prevention of cardiovascular disease, cancers, and diabetes, as well as probiotic, anti-inflammatory, and antimutagenic activities [11].

Rice flour contains ~90% total carbohydrates (with amylose content about 20%, ~47–90% starch, and ~2.4% dietary fiber of the total carbohydrate composition), ~6–10% protein (with dominant amino acids such as glutamine, aspartate, arginine, leucine, and valine) ~1–2% fat, and about 1% minerals [10,13]. Millet flour contains about 70% carbohydrates [14,15]. Starch is the main carbohydrate of millet grains (50–70% of the total carbohydrate composition), followed by a high content of dietary fiber (13.1% of the total carbohydrate composition) and a low content of monosaccharides (glucose, fructose, and galactose), disaccharide–sucrose, and trisaccharide–raffinose [15,16,17,18]. Millet flour contains about 9–12% proteins (with lysine as a dominant amino acid) and 1–2.6% oils (with dominant fatty acids such as palmitic, stearic, oleic, and linoleic) [15]. The importance of millet as a foodstuff, and therefore flour, is reflected in the content of biologically active components (vitamins B1 B2, B3, and E, and minerals K, Ca, P, Mg, Fe and Zn, as well as tannins, flavonoids, phenolic acids, and β-carotene) [15]. Regardless of the nutrition it possesses, the addition of millet flour can negatively affect the sensory quality of gluten-free products (small specific volume and increased hardness), thus the addition of a higher amount is often avoided [8].

However, the production of gluten-free bread is a technological challenge due to several drawbacks compared to gluten-rich bread, such as worst texture, less tasty, and lower nutritional quality due to a higher content in lipids and sugars, and a lower content in protein, dietary fiber, and mineral elements [7]. To overcome these problems, the use of nutritional valuable okara, a soy milk by-product, can be one of the solutions. Okara is characterized by a light-yellow color, mild and neutral flavor [19], and low energy potential (2.78–3.28 kcal/g fresh matter) [19,20]. From 1 kg of soybeans used in the manufacturing of soy milk, about 1.1–1.2 kg of okara is obtained [21,22]. The global production of soybean okara amounts to about 1.4 billion tons per year [23,24], but it is underutilized considering the potential nutritional benefits of okara, causing significant environmental pollution [25]. The composition of okara depends on the genotype of the soybean as well as the method of soy milk production [16,17,26,27,28]. Approximately 40% of produced okara is used for animal consumption and only 10% for human consumption; 50% of okara ends up as waste. Okara contains ~15–40% proteins (with essential amino acids: phenylalanine, leucine, isoleucine, lysine, valine, threonine, histidine and methionine, and non-essential amino acids: tyrosine, proline, alanine, arginine, glycine, glutamic acid, serine, and asparaginic acid) [29]. The dominant proteins in extracts of okara are subunits of basic 7S globulin (“heavy” (HI,II) and “light” (LI,II) subunits with molecular weight values of 27,000 and 16,000). Basic 7S globulin is desirable because of its nutritional value as it is a cysteine-rich glycoprotein [27]. Okara contains ~32–53% carbohydrates (with ~56–58% total dietary fibers and ~42–55% insoluble dietary fibers of the total carbohydrate composition and monosaccharides 0.17–4.11% (mannose and fructose) and disaccharides 1.61–4.35% (sucrose and maltose) depending on the soybean genotype [20]. In addition, okara contains lipids of a very wide range (~6–22%), depending on the genotype of the soybean and the method used to obtain soy milk (with dominant fatty acids: linoleic, oleic, palmitic, linoleic, and stearic fatty acids) [29]. Okara is rich in bioactive compounds (such as isoflavones and phytoestrogens; vitamins B1, B2, B3, B6, folates (B9), and K; antioxidants polyphenol isoquercetin) and minerals (Na, Mg, N, K, P, Ca, Fe, Zn, and Mn) [20,23,25,26,27,28,29,30,31]. The literatures data indicate that okara contains an antioxidant capacity similar to that of beet or pumpkin; much higher than those of tomato, carrot, broccoli, or onion [29]. Furthermore, okara lacks lactose, gluten, and cholesterol, which can have a significant health benefit for the health-compromised consumers [32].

The application of okara can have a significant effect on the structure of the gluten-free matrix and the volume of the bread. The high content of dietary fiber and protein can affect the rheological properties of bakery products [33]. In addition, dietary fibers are now often added to bakery products with the aim of prolonging freshness, which is based on their ability to retain water [34].

It has been shown that okara could be successfully used in the formulation of meat products, biscuits, filled pasta, drinks, candies, gluten bread, and nutritional flour, as well as edible packaging and biodegradable materials [21,25,35,36], due to its good nutritional and functional characteristics [37]. Guimarães et al. [38] formulated gluten-free bread with the addition of okara and corn by-products and obtained a product with improved nutritional characteristics, but poorer sensory/technological characteristics. However, the use of okara in the formulation of gluten-free bread (from different gluten-free grains and pseudocereals) has not yet been thoroughly investigated, whereby a product with good sensory properties has not yet been obtained.

Thus, the aim of this study was to produce soy okara-enriched gluten-free bread based on buckwheat, rice, and millet with high nutritional and low-energy values while achieving maximum sensory quality. Three gluten-free breads, each containing different contents of okara, and a control sample (without okara), were prepared and subjected to sensory evaluation. The bread with the highest sensory score was selected for further analysis of proximate composition and antioxidant properties. The new formulated bread can enable the diversification of gluten-free products on the market and increase a higher utilization of okara, food waste by-product, contributing to environmental protection.

## 2. Results and Discussion

### 2.1. Sensory Analyses

Knowing that gluten-free products are considered as products with poor sensory properties [38], sensory analysis of three gluten-free breads enriched with okara, and a control sample (without okara), was performed by first, aiming to select the most acceptable gluten-free bread for consumers. Preparation of sensory-acceptable bread for consumers is very demanding considering that the majority of consumers react negatively to the sensory characteristics of gluten-free bread, in addition to the unfavorable leguminous properties of soy food which are generally unacceptable for most consumers of the Western market [38,39]. The weighted mean scores for the sensory quality of the tested samples were relatively very similar, even for samples with 20% and 30% okara (4.30; Figure 1A), indicating that bread samples belong to the “very good quality” category. Regardless of the potential lipoxygenase activity, the tested breads had high taste scores. The highest score for taste (4.75) was given to the sample of bread with the highest percentage of okara (30%). This agrees with the results of the sensory analysis of soy bread, where it was concluded that beany flavor of soy-food was reduced, and soy flour contributed to a higher general acceptability of the bread [37].

It is very interesting that the bread containing 30% okara received a very high score for the shape (4.55), as well as for the marks related to the cross-section structure (4.20), considering that it does not contain gluten. Namely, from the technological aspect, the first problem that occurs in the production of gluten-free bread is the possible absence of desirable rheological and textural characteristics of the dough and the final product. Most common breads are made from wheat flour, water, salt, and yeast, with bread making relying on the ability of hydrated gluten to develop a viscoelastic network that traps gas (CO_2_) and produces bread with a larger loaf volume and better rheological and textural characteristics [7]. Our results are in contrast with the results of Ostermann-Porcel et al. [19] who identified that the greater presence of okara in gluten-free cookies caused a smaller volume, due to the fibers present in okara flour that interfere with the structure of the matrix, reducing the gas retention capacity of the dough. However, the same authors indicate that the results of scanning electron microscopy of the cross-section of cookies with and without okara showed no noticeable differences. The influence of soybean content on the volume of gluten-free dough/bread is opposite in the literature, too. For example, Melini et al. [40] examined the influence of soybean flour on the volume of gluten-free bread and indicated that there was an increase in the softness of bread with increasing soybean flour content. The rheological properties of the tested gluten-free bread can be influenced by many factors, such as the amount of water added during the preparation of the dough, the content of protein and dietary fiber, and the use of substances with a high affinity for hydrocolloid formation. The literatures data indicate that the presence of dietary fiber and hydrocolloids in optimal amounts can improve the texture of bread [38]. Moreover, the dominant proteins in extracts of okara are subunits of basic 7S globulin [27], which is a cysteine-rich glycoprotein [41] that can affect the structure of the gluten-free matrix by the formation of disulfide bridges.

In addition, bread with 30% okara received a high score for elasticity (4.20). This may be the effect of added guar gum to the dough, since it is known that all gluten-free breads contain hydrocolloids to improve dough behavior [42]. Namely, vegetable gums (such as guar gum) swell and form a gel, which thickens the mass of the dough preventing the loss of gas released during wetting and mixing. Hydrocolloids bind water, stabilize the structure of the crumb, and prevent rapid retrogradation of starch [43]. Additionally, the compatibility of rice flour protein (glutenin fraction in high concentration with albumin and globulin fraction) and soy protein (globulins–β conglycinin and glycinin) could be of particular importance for achieving good dough and bread elasticity. We can assume that the interaction between rice and soybean proteins in the dough was intensified by the direct formation of new, intermolecular covalent bonds, catalyzed by transglutaminases, and by the indirect formation of disulfide bonds. This combination of proteins from different sources and enzymes led to the formation of a network, which improves the structure of gluten-free breads. The interaction between rice and soy proteins in solutions was reported by Wang et al. [44]. The authors reported that protein composites showed significantly improved emulsifying and foaming properties compared to rise protein. Melini et al. [40] identified that the addition of soy flour to gluten-free dough/bread in the amount of 45–60% significantly increased softness.

Statistically significant differences were registered between the ratings of the sensory quality of the control sample and all samples enriched with okara, for all parameters of the sensory analysis, except for the smell. Among gluten-free bread samples, the sensory quality ratings between samples with 10% okara and samples with 20% and 30% okara differed for all quality parameters, except for the smell. Comparing the statistically significant differences for the sensory quality scores between the gluten-free bread enriched with 20% and 30% okara, the shape and taste scores were statistically different between the two samples.

In addition to similar mean quality scores, the samples differed according to the scores for individual parameters of sensory characteristics (Figure 1B). Gluten-free bread prepared with the addition of 30% okara by trained panelists received the highest scores for individual sensory characteristics; taste (4.75), odor (4.55), chewiness (4.15), and shape (4.55).

The obtained ratings (0–5) by consumers (Figure 2) confirmed the results of the sensory analysis obtained by trained expert panelists. After sensory analysis by consumers, the sample with the best evaluation of all quality parameters was the gluten-free bread enriched with 30% okara (Figure 2B). The mean quality rating of gluten-free bread with 30% okara was the highest (4.59) compared with other samples (Figure 2A), which introduced the product in the “excellent quality” category. Furthermore, the results that were obtained using a hedonic scale (1–9) showed a very high score of overall acceptability (8.71), which indicated a very pleasant/acceptable sensory feeling of gluten-free bread enriched with 30% okara.

Based on the obtained results of the sensory analysis, both trained evaluators, and consumers, rated the gluten-free bread enriched with 30% okara as the most acceptable for consumption. For this reason, this sample (with 30% okara) was selected as the most appropriate gluten-free bread enriched with okara for further analyses.

### 2.2. Proximate Composition

The proximate composition of gluten-free bread containing 30% okara is presented in Table 1. The high content of total carbohydrates was determined, 28.90% with high content of dietary fiber (14.00%, among which 11.11% were insoluble and 2.89% were soluble fibers). Pseudocereals and cereal are considered the most common sources of dietary fiber for bakery products, consisting of cellulose and complex xylans and lignin (in cell walls), arabinoxylans, β-glucan, heteromannans, and esterified phenolic acids (in aleurone and endosperm) [45]. However, legume fibers, including soybean, contain both insoluble (mainly consist of hemicelluloses and cellulose) and soluble dietary fibers (primarily consist of pectin) [46], and are considered to have more advantages than cereal fibers, due to higher content of solubles [45]. Taking this into account, soybean okara can increase the nutritional value of gluten-free bread.

Dietary fibers are poorly absorbable or non-absorbable in the human gastrointestinal tract and play a significant physiological/nutritional role in human metabolism [47]. Numerous studies prove their significant effect in preventing/treating various chronic diseases (such as diabetes, obesity, gastrointestinal tract, and cardiovascular diseases) and colorectal cancer [48,49,50,51,52]. Joint WHO and FAO experts recommend an intake of at least 25 g/day of dietary fibers, which can protect against obesity and its consequences [53]. Nevertheless, dietary fiber consumption by humans is usually lower than the recommended value [54]; thus, food technologists and food scientists are trying to develop fiber-enriched products. Bakery products, especially bread, are one of the most widely, and regularly consumed food worldwide [45], and can be an ideal source of dietary fiber and other bioactive compounds in a diet.

In addition to a high content of dietary fiber, gluten-free bread enriched with 30% okara has a very low sugar content (glucose, fructose, and sucrose) (0.05%; Table 1). Okara is characterized by a low content of monosaccharides and disaccharides, which influence the low value of total carbohydrates in a final product. Bearing in mind that these sugars significantly affect the glycemic index of food, it can be assumed that the analyzed bread is not characterized by a high glycemic index.

In addition to being a good source of dietary fiber, okara is also a good source of protein. The analyzed sample of gluten-free bread contained 8.80% of total proteins (Table 1). This is very important considering gluten-free bakery products generally have a lower content of total protein than similar products with gluten [55]. The reason for this is that gluten-free flours generally contain less protein than gluten-rich flours [7]. Therefore, people suffering from celiac disease, by consuming gluten-free bread, are forced to consume proteins from other sources. Segura and Rosell [56] investigated the composition of commercially available gluten-free breads and registered a total protein content of 0.91–2.80% in samples that did not contain added proteins (e.g., casein or proteins of lupine and egg). Therefore, the formulated gluten-free bread, according to legal regulations, can be considered a “product with increased protein content” [1].

Several important things must be considered when proteins are added to gluten-free bread. First, the functionality of added proteins is important since added proteins cannot replace the role of gluten in the dough. Second, the nutritional quality, allergenicity and origin (if it is a product intended for vegans), and price are very important aspects [7]. The addition of soy proteins required indication on the product declaration since they can be allergens in human nutrition. Alternately, it is very important that the allergenic effect of soy proteins can be reduced/neutralized by appropriate technological processes such as pressure techniques (e.g., extrusion and high hydrostatic pressure) and waves (e.g., gamma irradiation, microwave, and ultrasonication) [57].

Checking the possible presence of residual gluten in the formulated bread, 14.41 ppm was obtained. Considering that the FDA [58] prescribed that “any foods that carry the label “gluten-free,” “no gluten,” “free of gluten,” or “without gluten” must contain less than 20 parts per million (ppm) of gluten” this product can be classified in the group of “gluten-free products”.

The sample of gluten-free bread with 30% okara was characterized by a low content of total lipids (3.80%) and a very low content of saturated fatty acids (0.08%; Table 1), Such a low content of saturated fatty acids is of great nutritional importance, knowing that they can have a harmful effect on the human body (increase the content of total cholesterol and low-density lipoproteins, which leads to an increased risk of cardiovascular diseases) [59]. Considering that okara, depending on the method of production and soybean genotype, can contain total lipids in a very wide range (0.8–22%) [31], it can be assumed that in products enriched with okara, the content of total lipids will largely depend on their content in okara. For example, Ostermann-Porcel et al. [19] received gluten-free cookies enriched with 30% okara, which contained 16.80% of total lipids, significantly increased the total energy value of the product (387.6 kcal/100g). The total energy value of the formulated gluten-free bread enriched with 30% okara was very low (136.37 kcal/100g or 568.21 kJ/100g; Table 2), which potentially allows for the possibility to use this bread in the diets of obese people. The contribution of digestible carbohydrates to total energy value was the highest, more than 50%, Figure 3.

The content of table salt in the sample of gluten-free bread enriched with 30% okara was within the limits indicated in the literature (0.80%; Table 1). Šmídová and Rysová [43] reported that the salt content in gluten-free breads can vary widely, from 0.2% to as much as 2.5%. Sodium chloride in the technological sense plays a very important role as it extends the shelf life of the product (by ensuring the microbiological stability of the product), participates in the formation of the taste in food and in the formation of the dough structure of bakery products (where the addition of salt depends on the type of flour) [60]. However, low table salt content is desirable in food since high contents can lead to high blood pressure and cardiovascular problems [61].

Based on the obtained results, it can be concluded that the major drawbacks of commercially available gluten-free bread from nutritional quality aspects can be improved by the addition of 30% okara in the rice/buckwheat/millet flour mixture (Table 3). The results of this study confirm the statement of Melini et al. [40] which states that in the production of gluten-free bread, none of the raw materials can replace gluten, but legumes might “epresent a new forward-looking frontier in gluten-free breadmaking because of their functional and nutritional characteristics.”

### 2.3. Total Phenolic Content and Antioxidant Properties

The total phenolic content and antioxidant properties of gluten-free breads enriched with 30% okara are presented in Table 4. TPC in analyzed bread was 133.75 mgGAE/100g. In the literature available to us, there are no data on the content of total phenolics in gluten-free bread. However, in bread made from whole wheat flour, the content of total phenolics was ranged from 50 to 200 mgGAE/100g, depending on the cultivar of wheat [63]. Therefore, it can be concluded that the content of total phenolics in the prepared gluten-free bread was in the range of values for wheat bread.

The antioxidant properties of gluten-free bread were evaluated by three assays: FRP, ABTS, and DPPH. The reasoning is that different studies apply different methods and calculations of antioxidant properties; in addition, the combinations of different constituents in the formulation of bread are practically unlimited. In the case of complex food matrices, since there are many different potential antioxidants, of which have different mechanisms of action, it is recommended to use several different antioxidant screening assays [64]. The ABTS radical is soluble in water and organic solvents, enabling determination of hydrophilic and hydrophobic antioxidants, whereas the DPPH radical is soluble in organic solvents and is mainly used for the determination of phenolic antioxidants soluble in organic media [65]. Furthermore, the ABTS assay, unlike the DPPH assay which involves hydrogen atom transfer, are based on both hydrogen atom and electron transfer enabling determination of antioxidants with different mechanisms of action [65]. Antioxidants action involves the reduction of a colored ABTS/DPPH radical, whereas the FRP method monitors the reduction of ferric-iron (Fe^3+^) to ferrous-iron (Fe^2+^) by antioxidants [66]. All three methods indicated significant antioxidant activity (Table 4). The DPPH scavenging activity of the analyzed gluten-free bread was 49.42 mgTrolox/100g, whereas the ABTS radical cation scavenging activity was 86.80 mgTrolox/100g (Table 4). It was reported that the DPPH scavenging activity of bread prepared only with millet flour or rice flour was 19.24% and 10.50%, respectively [67]. There are no available data on the FRP of gluten-free bread, however, Shin et al. [37] estimated, using the FRAP method, that soy increases ferric reducing antioxidant power. Turfani et al. [68] indicated that the addition of leguminous vegetables provides bread with an acceptable volume, taste and texture, and increased antioxidant activity.

### 2.4. Macro- and Micro-Elements

Sodium (4677.08 µg/g) was registered as the most abundant macro-element in the gluten-free bread followed by phosphorus (2821.19 µg/g) and potassium (2002.04 µg/g), as well as sulfur (1374.72 µg/g). Additionally, the presence of calcium and magnesium were registered (849.59 µg/g; 777.04 µg/g; Table 5). Each of the registered macro-elements in the gluten-free bread enriched with 30% okara has an extremely important role in the body. Sodium is the main cation of the extracellular fluid and participates in the maintenance and regulation of the osmotic pressure of the blood plasma and other extracellular fluids, as well as in the synthesis of gastric HCl and in the regulation of the acid-base balance. Potassium, as the main intracellular cation, actively regulates osmotic pressure, affects the synthesis of proteins, ribosomes, and the activity of some enzymes (e.g., pyruvate kinase). In addition, potassium affects muscle activity, especially the activity of the heart muscle, and acts as an antagonist to calcium in its effect on heart rate. Calcium has a structural role in the body (part of solid tissues), neuromuscular (in muscle contraction, neurotransmitter release, and excitability control), enzymatic (as a coenzyme of blood coagulation factors), and hormonal (as an intracellular secondary messenger). Phosphorus participates in the process of ossification and deposition/transfer of energy as an integral part of macro-energetic compounds. Phosphorus is also involved in the reactions of phosphorylation and biosynthesis of some coenzymes. Magnesium ion is a cofactor and activator of many enzymes, participates in the activation of amino acids in protein synthesis, and reduces neuromuscular excitability. Sulfur is included in the structure of some amino acids and proteins, as well as in the thioester macro-energetic compound–coenzyme A and mucopolysaccharides. In addition, sulfur is involved in oxidation-reduction and detoxification processes in the body [69].

The most dominant micro-elements were iron (27.71 µg/g) and zinc (17.42 µg/g), but a significant content of manganese was also recorded (8.91). Iron as an integral part of specific proteins (heme proteins, e.g., hemoglobin, myoglobin, cytochromes, and proteins that do not contain heme but bind iron: ferritin, flavoproteins, transferrin) plays a central role in the transport of oxygen, as well as in the transport of electrons in the respiratory chain and participates in energy metabolism [66]. In addition, iron participates in the synthesis of steroid hormones and bile acids as well as in the detoxification of the organism [70]. Zinc is part of many enzymes, plays a role in the body’s immune system and significantly affects the preservation of cell integrity (by stabilizing the molecular structures of the components of cell membranes and cells). Manganese is an activator of many enzymes [70].

These results indicated a very favorable composition of macro- and micro-elements of the gluten-free bread. Ibidapo et al. [71] analyzed bread made from wheat flour and malted millet with 15% added okara and reported the highest content of calcium (1278.0 µg/g) among macro-elements, and iron (14.7 µg/g) and zinc (8.1 µg/g) among micro-elements. Maggio et al. [72] presented an overview of several traditional gluten-free products. Among other products, they demonstrated the mineral composition of gluten-free breadsticks that contained the highest content of sodium (3209.0 µg/g) and magnesium (1207.0 µg/g) among macro-elements, whereas iron (62.0 µg/g) was dominant among microelements. However, it is difficult to compare these results with the results of this study since the composition of the analyzed breadsticks is not known. Rybicka and Gliszczyńska-Świgło [73], after comparing the mineral composition of different gluten-free products concluded, that gluten-free products containing millet and buckwheat have a better composition of macro- and micro-elements than products containing rice.

The presence of potentially toxic elements was registered in analyzed samples: boron (5.82 µg/g) and aluminum (4.15 µg/g). Namely, boron plays a role in the synthesis of uridine-diphosphate-glucose, by affecting the reaction of glucose-1-phosphate into uridine-triphosphate and affects the activity of dehydrogenase and oxidoreductase enzymes. In larger amounts, boron will damage the brain. Aluminum is important in the activity of the nervous system, participates in cellular respiration, and activates B complex vitamins. However, aluminum can have a harmful effect on the central nervous system, kidneys, and bone marrow. For these reasons, food should always be checked for the presence of these potentially toxic elements [74]. The presence of aluminum was registered in gluten-free biscuits (18 µg/g) and breadsticks (1.5 µg/g) [72]. The presence of trace amounts of lead [72] which should not be present in food was also registered in these products. The presence of lead was not registered in the analyzed gluten-free bread.

### 2.5. Dietary Reference Intakes

The value of dietary reference intake showed the nutritional contribution of individual components of the analyzed gluten-free bread. A portion of 50 g of the analyzed gluten-free bread would meet as much as ~34% of the daily protein needs of children aged 1–3 years and ~23% of the needs of children aged 4–8 years. This is very important, considering that children are in the growing phase of life. Martos and López [74] obtained similar values (20%) for covering daily protein needs in children aged 4–8 years by a portion of gluten-free bread (50 g) enriched with *Prosopis nigra* flour. However, the same authors obtained lower values for the coverage of daily protein needs by examining gluten-free bread (portion 50 g) for men and females, which were 7–8%. The daily need for protein between men and females would be met by a portion of gluten-free bread (50 g) enriched with okara in the value of 9 to approximately 13% depending on age (Table 6).

A serving of gluten-free bread enriched with okara would meet the daily carbohydrate needs in the value of 11% for children, men, and females of all ages. However, the portion of analyzed gluten-free bread with okara would meet the daily needs in dietary fiber in a significantly higher percentage: for children aged 1–3 years ~37%, for children aged 4–8 years 28%, for men and females from 18 years to approximately 27%, depending on gender and age (Table 6). Such a high daily intake of dietary fiber is a consequence of the high presence of fiber in okara (~56–58% total dietary fibers of the total carbohydrate composition) [26]. The literature data show that a portion of gluten-free bread enriched with *Prosopis nigra* flour can contribute to the daily needs for dietary fiber in a significantly lower value (%DR = 10–16%) [74].

A portion of 50 g of analyzed gluten-free bread enriched with 30% okara would satisfy significant daily needs for macro- and micro-elements (Table 7 and Table 8).

### 2.6. Potential Benefits and Future Trends

In conclusion, there are several benefits of producing and consuming the formulated gluten-free bread enriched with 30% okara:

This bread does not contain gluten; thus, it is a suitable food for people who are intolerant to gluten and suffer from celiac disease. Studies have shown that 1% of the world’s population suffers from celiac disease and the only solution is to consume a gluten-free diet [76].

The high content of dietary fiber and the absence of sugar allow people with insulin resistance and diabetes to consume this bread. The range of products with these characteristics is significant since 442 million people in the world live with diabetes and that number is rapidly increasing [77]. By 2030, it is predicted that 7.8% of the world’s population will suffer from this disease [77]. While the prevalence of insulin resistance (metabolic syndrome) ranges from 15.5 to 46.5%, among adults, worldwide [78].

This bread is a source of nutritionally valuable proteins, which distinguish it from other gluten-free breads on the market and makes it suitable for athletes’ nutrition. For example, prolonged daily training can increase the need for proteins, not only because they encourage good muscle work but also regenerate damaged tissues [79].

A very low content of saturated fatty acids in the tested bread is desirable. It is known that different saturated fatty acids (short-chain, medium-chain, and long-chain) contribute differently to increasing the level of LDL cholesterol in the blood, and that a complex food system (which can significantly affect the digestion of lipids and their absorption) has a major impact [59,80]. Today, there are epidemiological data that saturated fatty acids can negatively affect coronary heart disease and the development of some types of cancer [81,82]. It is recommended to reduce the intake of saturated fatty acids to less than 10% of the total daily energy consumption and to replace them with unsaturated fatty acids [61].

The low energy value makes this bread suitable for use in diets for treating obesity. Worldwide, more than 1 billion people are obese—39 million children, 340 million adolescents, and 650 million adults—and these numbers are increasing [83]. WHO [83] estimates that “…by 2025, approximately 167 million people–adults and children–will become less healthy because they are overweight or obese”.

The presence of phenolics in the tested bread contributes to the antioxidant properties of this product. Thus, gluten-free bread enriched with 30% okara is a source of natural antioxidants, which is increasingly important in food preparation and nutrition. Namely, since about 1980, natural antioxidants have been used as an alternative to synthetic antioxidants. The toxicity of synthetic antioxidants was extensively studied indicating caution in their use. Therefore, in this sense, natural antioxidants represent nutritionally healthier and safer compounds compared to synthetic antioxidants [84,85].

Good sensory characteristics, above all good taste, provide the possibility of using this food by a wide group of consumers, who do not have metabolic and health problems, and want to eat healthy. Considering that bread is practically the most common food item in the diets of many people, and the low price of this product, it can be available to many consumers.

A potentially wide range of biologically active components (e.g., phenolics), minerals, and dietary fibers give this bread the properties of a “functional food”. The functional food sector in the food industry, with an annual growth rate of 7.4%, has experienced significant growth in recent years. The reason for such large growth in this area is not only due to technological progress in the food industry, but also due to the formulation of new products that will meet the needs of consumers who today are increasingly aware of the need to consume nutritionally/bioactive compound rich food to act preventively and preserve health [86].

Finally, but not in the least, the production of this bread is in accordance with the principles of a sustainable and circular economy, as well as procedures for proper waste handling in the food industry.

Regardless of the efforts made in the food industry, the production of gluten-free bread still presents significant technological problems. This confirms the poor quality of gluten-free bread currently available on the market [87,88]. The absence of gluten has a huge, negative impact on product characteristics and dough rheology, as the dough is less elastic, and the bread has a poorer texture and smaller volume compared to products containing wheat flour [87,88,89]. This is why it is a big challenge to obtain gluten-free bread by selecting and using raw materials that can replace/imitate the role of gluten in a good way, in order to obtain a product of satisfactory quality [90]. The results of this study indicate that using soy okara as an alternative source of protein can achieve these goals.

## 3. Material and Methods

### 3.1. Material

Buckwheat, rice, millet flours, sunflour oil, yeast, guar gum, and salt were purchased in the local markets. The soybean variety used in the production of okara was “Olga” selected by the Maize research Institute Zemin Polje (Belgrade, Serbia). Transglutaminase was provided by the Purtos Group in Serbia (Bijgaarden, Belgum).

### 3.2. Okara Production

Okara was obtained according to the procedure proposed by Stanojevic et al. [20], with modification. Soaked soybeans (grain:water = 1:5, 14 h, at 15 °C) were ground and cooked (grain:water = 1:6, 30 min, at 100 °C) in a soy milk maker (Mester, D1158-W11QG, Dongfeng Town, Zhongshan City, Guangdong Province, China). After filtering of the soybean suspension (through a muslin cloth) and hand squeezing, soy milk and okara were obtained. The okara was then dried (to a moisture content of 15%) on a heated surface with constant manual stirring.

### 3.3. Gluten-Free Bread Production

Gluten-free flour composed of buckwheat flour 45%, rice flour 33%, and millet flour 22% was mixed with okara in the ratios of 90%:10%; 80%:20%, 70%:30%, respectively. The ingredients necessary for quality dough were therefore added to the mixture of gluten-free flour and okara: vegetable oil (4%), yeast (4%), guar gum (0.03%), transglutaminase (0.01%), salt (1.8%), and water (160%). Sucrose was not added during the preparation of the dough. After preparing the dough and pouring it into molds, fermentation (at 25 °C; 30 min) and baking (at 175–180 °C; 55 min) of gluten-free bread were followed. The control bread was prepared in the same procedure but without adding okara.

### 3.4. Sensory Analyses

Sensory evaluation of bread was carried out 24 h after baking by 8 trained expert panelists, according to the procedure by Purić et al. [91] as well as by 150 consumers (79 women and 71 men). Evaluation of sensory qualities was performed with a point-system (0–5) by both groups of evaluators and with a “hedonic scale” by consumers. In the process of scoring with a point-system, weighted quality scores were made with the following importance of coefficients: for taste–6, for smell, chewiness, and all parameters of the structure of cross-section–3 and for shape–2. The maximum value of the mark during the evaluation was 5, so the quality category had five levels: excellent quality (quality score > 4.5), very good quality (3.5 < score ≤ 4.5), good quality (2.5 < score ≤ 3.5), poor/unsatisfactory quality (1.5 < score ≤ 2.5), and very poor quality (score ≤ 1.5) [92].

Using the “hedonic scale” in evaluating the overall acceptability of gluten-free bread enriched with 30% okara, consumers were required to give their opinion by choosing one of the 9 offered sensory feelings: (1 = extremely dislike, 2 = very much dislike, 3 = moderately dislike, 4 = slightly dislike, 5 = neither like/nor dislike, 6 = slightly like, 7 = moderately like, 8 = very much like, and 9 = extremely like) [92].

Evaluation with both the point system and the “hedonic scale” was performed in two repetitions. The consumers were non-smokers, and the sensory analysis was performed anonymously. None of the evaluators was informed about the composition of the samples before the evaluation. Samples were served on glass trays, with random number labels.

### 3.5. Proximate Composition

Total protein content was determined by Kjeldahl method [93] with a 6.25 as conversion factor. The content of insoluble and soluble dietary fiber was determined according to the AOAC 991.43 method [94]. The content of total carbohydrates was determined according by difference [95] and the sugar content (glucose, fructose, and sucrose) was determined using the SRPS E.L8.007:1980 and SRPS E.L8.011:1980 method [96,97]. Total lipid content was determined by Soxhlet method [98] and the content of saturated fatty acids was determined by SRPS EN ISO 12966-2:2017 [99]. The salt content was determined using the method of Julshamn and Lea [100]. For the quantification of gluten, immunological ELISA assay was used [101]. The content of macro- and micro-elements was determined using ICP-OES analysis according by Kostić et al. [102]; (instrument model Thermo Scientific iCAP 6500 Duo ICP, Thermo Fisher Scientific, Cambridge, United Kingdom; with iTEVA operating software). Total phenolic content (TPC) was determined according to the procedure described by Pešić et al. [103] and results were expressed as milligrams of gallic acid equivalents per 100 g of fresh weight (mgGAE/100g).

### 3.6. Total Energy Value

The total energy value was calculated from the proximate composition using the content of protein, lipid, and carbohydrate, multiplied by their combustion equivalents in the body (17 kJ/g for protein, 37 kJ/g for lipid, 17 kJ/g for available carbohydrate and 8 kJ/g for available dietary fiber) [97]. The results were expressed on a fresh weight basis.

### 3.7. Determination of Antioxidant Properties

The antioxidant properties of gluten-free bread enriched with okara was tested using three different tests: ferric reducing power assay (FRP), ABTS^•+^ and DPPH^•^ scavenging activity (ABTS/DPPH) according to Milinčić et al. [104]. The results of FRP assay were expressed as milligrams of ascorbic acid equivalents per 100 g of fresh weight (mg AA/100g). Antioxidant activity determined by the ABTS and DPPH assays were expressed as milligrams of Trolox equivalent per 100 g of fresh weight (mgTrolox/100g).

### 3.8. Dietary Reference Intakes

Percentage of the dietary reference intakes of nutrients (%DčRI) was calculated according to Martos and López [74] for children (from 1–8 years), men and women (from 9 to more than 70 years), and for pregnant and lactating. The calculation was made for a portion of bread of 50 g, based on the guidelines for Dietary Reference Intakes for energy, carbohydrate, fiber, fat, fatty acids, protein, and amino acids (macronutrients) by the National Academy of Science Institute of Medicine [75].

### 3.9. Statistical Analysis

Statistica software version 8.0 (StatSoft Co., Tulsa, OK, USA) was applied for statistical analysis. Data are expressed as mean and standard deviation of three replicates, or as mean and pooled standard deviation (*Pooled std*) of two replicates. In statistical data processing, Pearson’s correlation coefficients and Tukey’s test at *p* < 0.05 were applied.

## 4. Conclusions

Dried okara can be a great alternative as a new food ingredient. Okara is a suitable nutritional supplement in the production of gluten-free bread. According to current regulations, the formulated bread based on buckwheat, rice, and millet enriched with 30% okara belongs to the group of “gluten-free products” and “products with increased protein content”. The present research has shown that eco-innovative gluten-free bread with the addition (30%) of soy okara has the potential to be a source of carbohydrates, especially dietary fiber, high-value proteins, phenolic compounds, and macro- and micro-elements that are valuable in human health. The analyzed bread showed good antioxidant properties and high sensory scores, obtained for taste, shape, odor, chewiness, and cross-section properties. Gluten-free bread enriched with 30% okara can potentially meet significant values of the daily needs for carbohydrates, dietary fiber, protein, macro- and micro-elements of children, men, and females. In addition to its high nutritional value, this bread was distinguished by its low energy value, thus it can be suitable in diets intended for the treatment of obesity.

In addition, the production of this eco-innovative bread was in accordance with the sustainable method of production, circular economy, and proper waste management in food production, which is one of the important goals of modern life from an economic and ecological aspect.

## Figures and Tables

**Figure 1 molecules-28-04098-f001:**
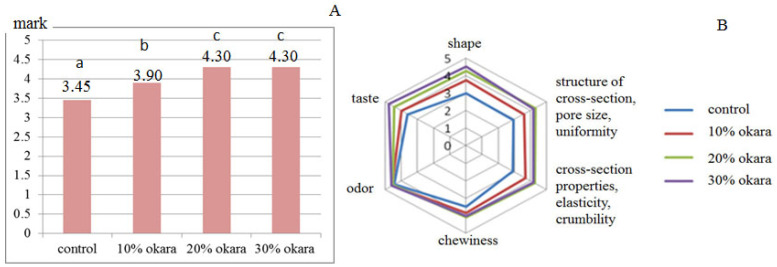
Sensory analysis by trained expert panelists of the bread samples supplemented with 10%, 20%, and 30% okara, respectively, compared with control bread after 24 h. (**A**)—values of weighted scores; (**B**)—scores of individual sensory quality parameters. Means with different small roman letters in Figure 2A are significantly different (*p* < 0.05).

**Figure 2 molecules-28-04098-f002:**
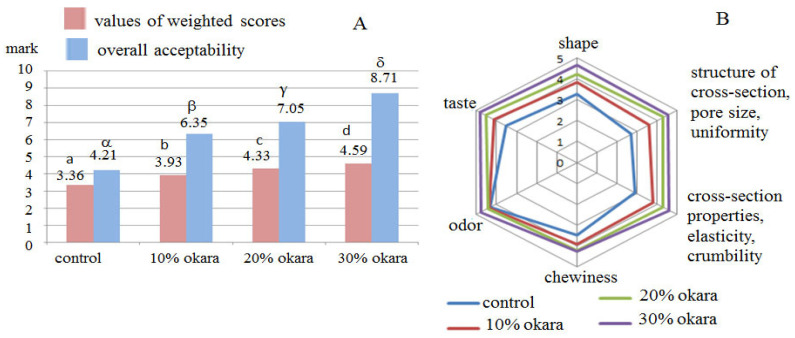
Sensory analysis by consumers of the bread samples enriched with 10%, 20%, and 30% okara, respectively, compared with control bread after 24 h. (**A**)—values of weighted scores (0–5) and overall acceptability obtained by hedonic scale (1–9); (**B**)—scores of individual sensory quality parameters (0–5). Means with different small roman and greek letters in Figure 2A are significantly different (*p* < 0.05).

**Figure 3 molecules-28-04098-f003:**
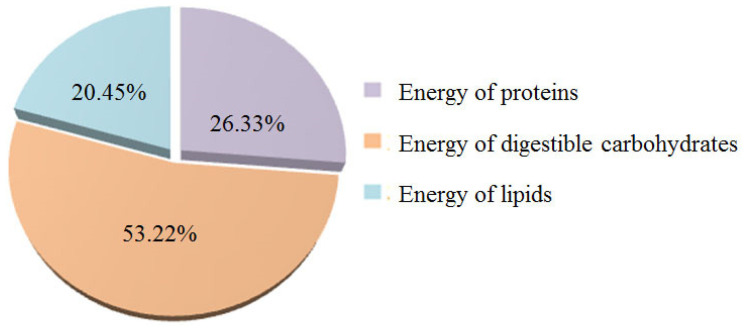
Percentage participation of individual components in the total energy value of gluten-free bread enriched with 30% okara.

**Table 1 molecules-28-04098-t001:** Proximate composition of gluten-free bread enriched with 30% okara.

Components	Content (%)
Total proteins	8.80 ± 0.07
Total carbohydrates	28.90 ± 0.04
Insoluble fiber	11.11 ± 0.03
Soluble fiber	2.89 ± 0.06
Sugars (glucose, fructose, sucrose)	0.05 ± 0.006
Total lipids	3.80 ± 0.03
Saturated fatty acids	0.80 ± 0.04
Salt	0.80 ± 0.03
	Content (ppm)
Gluten	14.41 ± 5.33

All results were calculated based on the dry weight of the sample.

**Table 2 molecules-28-04098-t002:** Energy value of gluten-free bread enriched with 30% okara.

Components	kJ/100g	kcal/100g
Energy of proteins	149.60	35.91
Energy of digestible carbohydrates *	302.43	72.58
Energy of lipids	116.18	27.88
Total energy value	568.21	136.37

* The content of digestible carbohydrates was calculated from the difference between total carbohydrates and insoluble dietary fibers.

**Table 3 molecules-28-04098-t003:** Nutritional quality problems of gluten-free products suggested by Gómez [7] and possible solutions to address them by gluten-free bread enriched with 30% okara.

Nutritional Aspects to Improve	Solution	Outcome
Low protein	Okara proteins are added	High-quality proteins were added and the resulting bread, according to legal regulations, belongs to the group “product with increased protein content”. Okara proteins are of high value and contain all essential amino acids, with lysine exceeding the daily requirements. Furthermore, they can reduce the content of cholesterol and triglycerides in the blood. Such properties allow okara proteins to be used for supplementation [19].
Low fiber	Okara fibers are added	Soluble and insoluble fibers are added with okara (dietary fibers of formulated gluten-free bread make up to 50% of total carbohydrates)
Low minerals	Okara minerals are added	Okara contains: Na, Mg, N, K, P, Ca, Fe, Zn, Mn; okara contain Fe^+2^, which is easily absorbed
High lipids	Okara may contain a low percentage of total lipids [28].	The addition of okara did not increase the lipid content in the formulated gluten-free bread; in addition, the saturated fatty acid content was low. Namely, soy fats, are known as “cardio-healthy fats”, they are unsaturated fats (contains an essential omega 3 fatty acids) [62].
High sugars	Okara contain a low percentage of monosaccharides and disaccharides [20].	The content of sugar (glucose, fructose, and sucrose) in the examined bread was very low.

**Table 4 molecules-28-04098-t004:** Total phenolic content and antioxidant properties of gluten-free bread enriched with 30% okara.

Total Phenolics	FRP	ABTS	DPPH
mgGAE/100g	mgAA/100g	mgTrolox/100g	
133.75 ± 2.42	119.25 ± 4.29	86.80 ± 2.46	49.42 ± 8.58

All results were calculated based on the fresh weight of the sample. FRP-ferric reducing power. assay; ABTS and DPPH radical scavenging activity assay.

**Table 5 molecules-28-04098-t005:** Content of mineral elements of gluten-free bread enriched with 30% okara.

Macro-Elements (µg/g)			
Ca	849.59	Na	4677.08
K	2002.04	P	3831.19
Mg	777.04	S	1374.72
Pooled std	4.11		
Micro-elements (µg/g)			
Co	n.d.	Mn	8.91
Cr	0.19	Ni	0.87
Cu	2.96	Sr	2.42
Fe	24.71	Zn	17.42
Pooled std	0.25		
Toxic elements (µg/g)			
Al	4.15	Cd	0.02
As	n.d.	Li	0.04
B	5.82	Pb	n.d.
Ba	0.59		
Pooled std	0.09	Pooled std	0.00

Data are expressed as mean and pooled standard deviation (Pooled std) of two replicates. n.d. <0.005 µg/g.

**Table 6 molecules-28-04098-t006:** Dietary reference intakes (%DRI) for carbohydrate and protein covered by a portion of 50g of gluten-free bread enriched with 30% okara.

Life Stage Group	Nutrients								
	Carbohydrate			Total Fiber			Proteins		
	DRI (g/d)	CBS (g/50g)	%DRI	DRI (g/d)	CBS (g/50g)	%DRI	DRI (g/d)	CBS (g/50g)	%DRI
Children		14.45	11.12		7.00			4.40	
1–3 y	130			16		36.84	13		33.85
4–8 y				25		28.00	19		23.16
Man									
9–13 y				31		22.58	34		12.94
14–18 y				38		18.42	52		8.46
19–50 y									
51–70 y				30		23.33	56		7.86
>70 y									
Females									
9–13 y				26		26.93	34		12.94
14–18 y							46		9.57
19–70 y				21		33.33			
>70 y									
Pregnancy	175		8.26	28		25.00	71		6.20
Lactation	210		7.19	29		24.14			

DRI-Dietary Reference Intakes [75]. No values for fat were detected, thus %RDI values for a portion of the tested bread are not presented in this study. CBS-Contribution of a bread serving (50 g). d—day; y—year.

**Table 7 molecules-28-04098-t007:** Dietary reference intakes (%DRI) for macro-elements covered by a portion of 50 g of gluten-free bread enriched with 30% okara.

Life Stage Group	Macro-Elements					
	Ca	K	Mg	Na	P	S
Children						
1–3 y	93.04	3.34	48.57	33.29	41.64	68.74
4–8 y	58.15	2.63	29.89	19.52	38.31	
Men						
9–13 y	35.78	2.22	16.19	15.53	15.32	68.74
14–18 y		2.13	9.48			13.75
19–30 y	46.52		9.71		27.37	
31–50y			9.25			
51–70 y	38.77			17.92		
>70 y				19.42		
Females						
9–13 y	35.78	2.22	16.19	17.92	15.32	68.74
14–18 y		2.13	10.79			13.75
19–30 y	46.52		12.53		27.37	
31–50 y			12.14			
51–70 y	38.77			19,42		
>70 y						
Pregnancy	46.52	2.13	11.10	15.53	27.37	/
Lactation		1.96	12.34			

y—year.

**Table 8 molecules-28-04098-t008:** Dietary reference intakes (%DRI) for micro-elements covered by a portion of 50 g of gluten-free bread enriched with 30% okara.

Life Stage Group	Micro-Elements					
	Fe	Zn	Mn	Cu	Cr	Ni
Children						
1–3 y	17.65	28.67	37.13	43.53	86.36	21.75
4–8 y	12.36	17.20	29.70	33.64	63.33	14.50
Men						
9–13 y	15.44	10.75	23.45	21.14	38.00	7.25
14–18 y	11.23	7.82	20.25	16.63	27.14	4.35
19–50 y	15.44		19.37	16.44		
51–70 y					31.67	
>70 y						
Females						
9–13 y	15.44	10.75	27.84	21.14	45.24	7.25
14–18 y	8.24	9.56	24.75	16.63	39.58	4.35
19–50 y	6.86	10.75	22.28	16.44	38.00	
51–70 y	15.44				47.50	
>70 y						
Pregnancy	4.58	7.82	22.28	14.80	31.67	
Lactation	13.73	7.17	17.13	11.38	21.11	

y—year.

## Data Availability

Data are contained within the article.

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
