# Peer review of "Okara-Enriched Gluten-Free Bread: Nutritional, Antioxidant and Sensory Properties"

_molecules, 2023, doi:10.3390/molecules28104098_

Round 1

Reviewer 1 Report

This paper dealt with Okara-enriched gluten-free bread: nutritional, antioxidant and sensory properties. The authors need to address the questions raised.

Abstract: 

~Three gluten-free bread with different content of okara was prepared and subjected to 16 sensory evaluation.~ Q: In addition to sensory evaluation, please add other methods used in the study in this sentence.  

Q: Do you do a control test? 

Q: What are the results of three gluten-free bread and the differences between them? 

Q: What are the formulations of three gluten-free bread? The authors need to address them in the abstract. 

Introduction:

Q: Please provide more information regarding Okara.

Table 1-3, 5-8 and figure 2: 

Q: Why did the authors show the results of just one formulation. How about the other formulations? 

Table 4: 

Q: Is table 4 based on the results of this study? The authors need to address clearly.

Table 9 and 10:

Q: How to determine the dietary reference intakes? Can they be obtained from this study?

Author Response

This paper dealt with Okara-enriched gluten-free bread: nutritional, antioxidant and sensory properties. The authors need to address the questions raised.

Thank you very much for your comments and suggestions. The required corrections have been made and all issues are addressed below.

Gómez, M. (2022). Chapter Five - Gluten-free bakery products: Ingredients and processes. In W. Zhou & J. Gao (Eds.), Advances in Food and Nutrition Research (Vol. 99, pp. 189-238). Academic Press. https://doi.org/https://doi.org/10.1016/bs.afnr.2021.11.005

Guimarães, R. M., Pimentel, T. C., de Rezende, T. A. M., Silva, J. d. S., Falcão, H. G., Ida, E. I., & Egea, M. B. (2019). Gluten-free bread: effect of soy and corn co-products on the quality parameters. European Food Research and Technology, 245(7), 1365-1376. https://doi.org/10.1007/s00217-019-03261-9

Ioniță-Mîndrican, C. B., Ziani, K., Mititelu, M., Oprea, E., Neacșu, S. M., Moroșan, E., Dumitrescu, D. E., Roșca, A. C., Drăgănescu, D., & Negrei, C. (2022). Therapeutic Benefits and Dietary Restrictions of Fiber Intake: A State of the Art Review. Nutrients, 14(13). https://doi.org/10.3390/nu14132641

Magano, N., du Rand, G., & de Kock, H. (2022). Perception of Gluten-Free Bread as Influenced by Information and Health and Taste Attitudes of Millennials. Foods, 11(4), 491. https://www.mdpi.com/2304-8158/11/4/491 

Milinčić, D. D., Stanisavljević, N. S., Kostić, A. Ž., Gašić, U. M., Stanojević, S. P., Tešić, Ž. L., & Pešić, M. B. (2022). Bioaccessibility of Phenolic Compounds and Antioxidant Properties of Goat-Milk Powder Fortified with Grape-Pomace-Seed Extract after In Vitro Gastrointestinal Digestion. Antioxidants, 11(11), 2164. https://doi.org/doi:10.3390/antiox11112164

Moore, M. M., Schober, T. J., Dockery, P., & Arendt, E. K. (2004). Textural Comparisons of Gluten-Free and Wheat-Based Doughs, Batters, and Breads. Cereal Chemistry, 81(5), 567-575. https://doi.org/https://doi.org/10.1094/CCHEM.2004.81.5.567

Sandri, L. T. B., Santos, F. G., Fratelli, C., & Capriles, V. D. (2017). Development of gluten-free bread formulations containing whole chia flour with acceptable sensory properties. Food Science & Nutrition, 5(5), 1021-1028. https://doi.org/https://doi.org/10.1002/fsn3.495

Šmídová, Z., & Rysová, J. (2022). Gluten-Free Bread and Bakery Products Technology. Foods, 11(3), 480. https://www.mdpi.com/2304-8158/11/3/480 

Stanojevic, S. P., Barac, M. B., Pesic, M. B., Zilic, S. M., Kresovic, M. M., & Vucelic-Radovic, B. V. (2014). Mineral elements, lipoxygenase activity, and antioxidant capacity of okara as a byproduct in hydrothermal processing of soy milk  [Article]. Journal of Agricultural and Food Chemistry, 62(36), 9017-9023. https://doi.org/10.1021/jf501800s 

Abstract: 

~Three gluten-free bread with different content of okara was prepared and subjected to 16 sensory evaluation.~ Q: In addition to sensory evaluation, please add other methods used in the study in this sentence.  

Thank you very much for your suggestion. The Abstract is rewritten according to suggestion of all reviewers. The methods are listed in lines 21-23.

Q: Do you do a control test? 

Yes, the control sample, without okara, is used for sensory analysis. For clarification, the explanation is added in the abstract, line 18, material and methods, lines 597-598, results and discussion, lines 138-139.

Q: What are the results of three gluten-free bread and the differences between them? 

The differences between the 3 gluten-free bread with different ratio of okara are explained in the sensory evaluations section and is shown in Figure 1. Analyzes of physicochemical and functional properties were performed for the best sensory-rated sample.

Q: What are the formulations of three gluten-free bread? The authors need to address them in the abstract. 

Thank you for the suggestion. The required information is added. See lines 16-18. A detailed description of the bread preparation was already given in the Material and Methods section.

Introduction:

Q: Please provide more information regarding Okara.

Thank you for the suggestion. The additional information is added in the introduction section. See lines 94-112.

Table 1-3, 5-8 and figure 2: 

Q: Why did the authors show the results of just one formulation. How about the other formulations? 

The primary goal of this research was to prepare sensory acceptable bread for consumers. This task is very demanding considering that the majority of consumers react negatively to the sensory characteristics of gluten-free bread as well as to the unfavorable leguminous properties of soy-food, which are generally unacceptable for most consumers of the Western market (Guimarães et al., 2019; Sandri et al., 2017).  

So the work plan was to formulate gluten-free bread acceptable for the consumers and then conduct physicochemical and functional property analysis only for the sensory most acceptable bread.

For clarification, additional sentences were added in the section Results and discussion, lines 137-140; 228-230                .

Table 4: 

Q: Is table 4 based on the results of this study? The authors need to address clearly.

Thank you for this observation. We regret that the presentation of Table 4 has caused confusion. Namely, the explanations in Table 4 are given on the basis of the results obtained in this study, with the aim to address the problems in the quality of gluten-free bread defined by  (Gómez, 2022). The title of the Table 4 was changed to avoid confusion, line 351-352.

Table 9 and 10:

Q: How to determine the dietary reference intakes? Can they be obtained from this study?

The dietary reference intakes were calculated from the results of this study (tables 8, 9, and 10) as the percentage of daily needs of certain nutrients after consuming 50g of the gluten-free bread enriched with 30% okara based on the DRI values prescribed by the National Academy of Science Institute of Medicine.

In our earlier research, a similar calculation was done for the daily intake of microelements with the consumption of 100g of biscuits containing okara (Stanojevic et al., 2014).  

Reviewer 2 Report

Line 1 was selected for further analysis. The formulated gluten-free bread is characterized by a 18.

Please, change is characterized to was characterized

Line 53 sucrose and trisaccharide - raffinose [11-14]. Millet flour contain about 9-12% proteins 53

Please, change contain to contains

Line 99 The highest score for taste (4.75) was given to the sample of bread with the highest percentage of okara (30%).

They are statistically significant differences. Indicate if so the significance. In addition, the figures should indicate which samples and for which sensory parameters the differences are significant.

Table 1 expresses significant differences, which samples are compared, the bread with 30% okara versus the control?

Line 168 Remove tale 2

From my point of view, correlation studies between chemical composition parameters do not make sense and table 2 should be removed

Line 436 The analyzed bread was characterized by good anti- oxidant activity and high ratings of sensory attributes.

General comments

In my opinion, the article is interesting, but it would gain greatly if the sensory characteristics of gluten-free bread were compared with gluten-containing bread and if the physicochemical characteristics and antioxidant capacity of bread with different percentages of okara were also compared. I do not understand why the analyzes of the breads are limited to the best valued sensorially, also taking into account that an acceptance test has not been carried out by consumer judges. Taking into account that the incorporation of okara seems to improve the properties of the breads and it is a by-product, why limit it to 30%?

Why hasn't the sensory quality of gluten-free breads been compared to a gluten-containing bread?

To decide which gluten-free bread formula is the most appropriate, a sensory test with trained judges is not the best option, but to describe the products .A consumer acceptance test should be carried out (untrained judges and with a number of panelists not less than 100).

Was the sensory evaluation of the breas by the trained judges performed under blind conditions, without information about the composition of the breads, or under informed conditions?

How were sensory parameters measured, structured or unstructured scales? and what was the range of scores (0-9), (0-5)?

Author Response

Thank you very much for your comments and suggestions. The required corrections have been made and all issues are addressed below.

Line 1 was selected for further analysis. The formulated gluten-free bread is characterized by a 18.

Please, change is characterized to was characterized

Thank you very much for this remark. The required correction was done, line 27.

Line 53 sucrose and trisaccharide - raffinose [11-14]. Millet flour contain about 9-12% proteins 53

Please, change contain to contains

Thank you very much for this remark. The required correction was done, line 72

Line 99 The highest score for taste (4.75) was given to the sample of bread with the highest percentage of okara (30%).

They are statistically significant differences. Indicate if so the significance. In addition, the figures should indicate which samples and for which sensory parameters the differences are significant.

Thank you very much for this suggestion. An explanation of the statistically significant differences between the scores for the sensory quality of the samples was added, lines 206-213.

Table 1 expresses significant differences, which samples are compared, the bread with 30% okara versus the control?

Thank you very much for this observation. It was an unintentional mistake. In Table 1, small letters are deleted as well as comment.

Line 168 Remove tale 2

From my point of view, correlation studies between chemical composition parameters do not make sense and table 2 should be removed

Thank you very much for the comment. Table 2 and referring text were removed from the text.

Line 436 The analyzed bread was characterized by good anti-oxidant activity and high ratings of sensory attributes.

Thank you for the suggestion. The sentence is reworded, lines 566-567.

 General comments

In my opinion, the article is interesting, but it would gain greatly if the sensory characteristics of gluten-free bread were compared with gluten-containing bread and if the physicochemical characteristics and antioxidant capacity of bread with different percentages of okara were also compared. I do not understand why the analyzes of the breads are limited to the best valued sensorially, also taking into account that an acceptance test has not been carried out by consumer judges. Taking into account that the incorporation of okara seems to improve the properties of the breads and it is a by-product, why limit it to 30%?

Thank you for your opinion and issues raised. The answers are listed above.

It would be difficult to compare the characteristics of gluten and gluten-free bread, especially from the aspect of sensory and rheological properties, because the "winner" would always be gluten-containing bread due to the fact that gluten is the main structure-forming protein in wheat flour, responsible for the elastic and extensible properties needed to produce good quality bread (Moore et al., 2004; Sandri et al., 2017). However, thank you for your suggestion, the comparison of gluten-containing and gluten-free bread will be considered in future research.

The major obstacle to the acceptance of gluten-free bread by consumers is its sensory properties. This opinion is supported by the statement of Sandri et al. (2017) "Despite the considerable advances in gluten-free (GF) research and the impressive growth of the GF market in recent years, individuals with gluten-related disorders still have trouble finding GF products because of high prices, limited variety and availability and poor sensory properties. That's why it was important to first formulate a product with good sensory properties and then did the product’s physicochemical and functional characterization.

Thank you for your suggestion to test consumers’ acceptance of formulated gluten-free bread. The test of sensorial quality and overall acceptability by consumers was performed and results were incorporated into the text. See lines: 219-227

The addition of okara in gluten-free bread significantly increases the content of dietary fiber. Consumption of 50g of formulated bread meets from 27% to 37% of the daily needs in dietary fiber depending on gender and age. The excessive addition of dietary fiber to the diet can lead to side effects including gas, increased abdominal pain, and possible malabsorption of some nutrients (Ioniță-Mîndrican et al., 2022). Thus, this is the reason why the addition of okara was limited to 30%.

Why hasn't the sensory quality of gluten-free breads been compared to a gluten-containing bread?

The sensory properties of standard wheat bread are preferred over gluten-free bread (Magano et al., 2022). Due to it, consumers of gluten-free bread are people who are intolerant to gluten or suffer from celiac disease. This is the reason why sensory properties were not performed with gluten-containing bread. However, the comparison between the sensory properties of gluten-containing and gluten-free bread will be considered in future research.

To decide which gluten-free bread formula is the most appropriate, a sensory test with trained judges is not the best option, but to describe the products. A consumer acceptance test should be carried out (untrained judges and with a number of panelists not less than 100).

Thank you very much for this suggestion. The required test was performed and included into the text. See lines 219-227; 601-604;610-619. 

Was the sensory evaluation of the breas by the trained judges performed under blind conditions, without information about the composition of the breads, or under informed conditions?

Thank you very much for this observation. Sensory analysis by trained judges was performed in "blind" conditions without information about the composition of the samples they are evaluating. Sensory analysis by consumers was also done the same. An additional explanation was added to the text. See lines 615-619.

How were sensory parameters measured, structured or unstructured scales? and what was the range of scores (0-9), (0-5)?

Thank you very much for this observation. Trained evaluators performed the sensory evaluation using a point system (0-5). Consumers made a sensory assessment using the points of the system (0-5) and the hedonic scale (1-9). Additional explanations were added to the text. See lines 601-604; 610-619.

Ioniță-Mîndrican, C. B., Ziani, K., Mititelu, M., Oprea, E., Neacșu, S. M., Moroșan, E., Dumitrescu, D. E., Roșca, A. C., Drăgănescu, D., & Negrei, C. (2022). Therapeutic Benefits and Dietary Restrictions of Fiber Intake: A State of the Art Review. Nutrients, 14(13). https://doi.org/10.3390/nu14132641

Magano, N., du Rand, G., & de Kock, H. (2022). Perception of Gluten-Free Bread as Influenced by Information and Health and Taste Attitudes of Millennials. Foods, 11(4), 491. https://www.mdpi.com/2304-8158/11/4/491 

Moore, M. M., Schober, T. J., Dockery, P., & Arendt, E. K. (2004). Textural Comparisons of Gluten-Free and Wheat-Based Doughs, Batters, and Breads. Cereal Chemistry, 81(5), 567-575. https://doi.org/https://doi.org/10.1094/CCHEM.2004.81.5.567

Sandri, L. T. B., Santos, F. G., Fratelli, C., & Capriles, V. D. (2017). Development of gluten-free bread formulations containing whole chia flour with acceptable sensory properties. Food Science & Nutrition, 5(5), 1021-1028. https://doi.org/https://doi.org/10.1002/fsn3.495

Reviewer 3 Report

I am very grateful you for the invitation to review manuscript molecules-2344077 by Pešić and coauthors "Okara-enriched gluten-free bread: nutritional, antioxidant and sensory properties”. The aim of this study was to produce an eco-innovative gluten-free bread with a pleasant taste and a unique formulation that includes the highest quality grains and pseudocereals (buckwheat, rice, and millet) and okara, a by-product of soy milk production. The work is interesting but needs adjustments to increase the quality of the material.

Comments:

- Abstract, Line 13: Please review the “eco-innovative” designation, as several criteria must be taken into account to indicate a product in this category.

- Line 16: Indicate proportions used.

- Abstract: Please indicate in the abstract a brief and better step-by-step about the work and the evaluated parameters.

- Lines 17-18: Indicate the chosen formulation.

- Line 18: Change “The formulated gluten-free bread is characterized” to “The formulated gluten-free bread was characterized”.

- Line 14 “High sensory scores”: Indicate numerical values.

- Lines 26-27: Please replace with a more accurate conclusion of the work.

- Lines 32-34: Explain better the issue related to celiac disease. Check other information at https://doi.org/10.1007/s11947-022-02975-1

- Lines 40-44: Indicate the benefits of consumption of these components.

- Line 69: Check the term “very low”, since more than 50% of the okara is reused, a much higher amount than other by-products.

- Introduction: Include a brief technological explanation regarding the role of ingredients and the challenges of substitution in gluten-free matrices.

- Lines 72-77: Indicate textually what explains such a large variation in the components.

- lines 82-83: Please review the sentence, as some products have already been prepared with the addition of okara. Indicate the differential in relation to the current study (check the excellent explanation of lines 84-91):

10.1007/s00217-019-03261-9

10.15406/mojfpt.2017.04.00111

And others

- Line 86: Change “conent” to “content”.

- 4. Material and methods: Check the correct order of presentation of the methodology. Other steps were performed before the sensory analysis of the bread.

- Line 462: Please check the correct citation form.

- Lines 469-470: Check the sentence, as some information about the procedure is missing o.

- Line 473: salt and water (160%). Is this information correct?

- Lines 113-130: Add possible explanations of matrices or other steps that can justify the opposite of what is verified in the literature for the influence of okara in the matrix of bread.

-  Lines 139-144: Describe whether this has been verified in other works.

- Line 156: Check the correct writing of the sentence “parameters; Table 2).”

- Lines 207-208: Specify better what corresponds to the appropriate technological processes.

- Line 229: Check the sentence “carbohydrates, Figure 2)”.

- Lines245-248: Please check the sentence, applied salt concentration is low for microbial inhibition.

- Change “FRP” to “FRAP” derived from Ferric Reducing Antioxidant Power Assay”.

- Lines 275-277: Better explain the use of different methods, which are based on the characteristics of the various components, such as polarities, and categories, among others.

- Antioxidant activity: Please compare with synthetic antioxidants widely reported in the literature.

- 2.4. Macro- and microelements: improve the discussion based on the role of minerals in the body or technological function (rather than concentration comparison with other work).

- Lines 359-364: And how can this be resolved?

Author Response

I am very grateful you for the invitation to review manuscript molecules-2344077 by Pešić and coauthors "Okara-enriched gluten-free bread: nutritional, antioxidant and sensory properties”. The aim of this study was to produce an eco-innovative gluten-free bread with a pleasant taste and a unique formulation that includes the highest quality grains and pseudocereals (buckwheat, rice, and millet) and okara, a by-product of soy milk production. The work is interesting but needs adjustments to increase the quality of the material.

Thank you very much for your opinion as well as for all suggestions to improve the quality of the manuscript. All suggestions were accepted and adequately addressed.

Comments:

- Abstract, Line 13: Please review the “eco-innovative” designation, as several criteria must be taken into account to indicate a product in this category.

Thank you very much for this suggestion.  The term "eco-innovative" is relatively recently introduced and arose as a result of people's awareness of the importance of protecting the environment. According to the definition of eco-innovation given by the EU project "SUPER project” granted by the European Regional Development Fund https://projects2014-2020.interregeurope.eu/super/news/news-article/373/superproject-what-exactly-does-eco-innovation-mean/ as follow:

 "eco-innovative/product/action as follows: "Eco-innovation could be as well alternative product as venture concept (introduction of a new method of production), a new component of a production system or a new architecture for that system. It may also be reflected in the conquest of a new source of supply of raw materials or semi-manufactured goods, more efficient or the new way of its exploitation and use. It could be recognized also in the carrying out of the new organization of any industry, focusing on environmental responsibility. Sustainable eco-innovation includes a large range of activities related to among others to areas like: water supply, waste management and remediation activities, bio-economy, green energy, biotechnology (molecular technologies for medicine and biopharmacy), food security and food technologies (functional food; bioresource)."

https://projects2014-2020.interregeurope.eu/super/news/news-article/373/superproject-what-exactly-does-eco-innovation-mean/

the gluten-free bread enriched with okara belongs to eco-innovative product based on several findings:

  1. "..alternative product as venture concept (introduction of a new method of production)..." - there is insufficient data on the production of gluten-free bread with the addition of okara, so this product can be considered as a "new method of production";

  1. "...a new component of a production system..." - okara represents a new component in the gluten-free bread production system;

  1. "...It may also be reflected in the conquest of a new source of supply of raw materials or semi-manufactured goods, more efficient or the new way of its exploitation and use..". - the production of gluten-free bread with okara represents a new way of efficient use of okara (which is a by-product in the soy-food industry;

  1. "...It could be recognized also in the carrying out of the new organization of any industry, focusing on environmental responsibility." - This method of producing gluten-free bread with the addition of okara belongs to the food industry with a significant focus on reducing organic waste (okara) and its sustainable treatment, which is of great importance in the pursuit of maintaining a healthy environment;
  2. "...Sustainable eco-innovation includes a wide range of activities related, among other things, to areas such as: ... waste management ..., food safety and food technology (functional food...) - we believe that in In this research, the gluten-free bread obtained/characterized is a product created by applying the principles of sustainable waste management in food technology, and it can be concluded (on the basis of the nutritional characteristics of gluten-free bread with 30% okara) that a functional food has been obtained.

- Line 16: Indicate proportions used.

Thank you very much for this suggestion. The required data was provided. See lines 16-17.

- Abstract: Please indicate in the abstract a brief and better step-by-step about the work and the evaluated parameters.

Thank you very much for this suggestion. The abstract was rewritten as you suggested.

- Lines 17-18: Indicate the chosen formulation.

Thank you very much for this suggestion. The required data was provided. See lines 23-26.

- Line 18: Change “The formulated gluten-free bread is characterized” to “The formulated gluten-free bread was characterized”.

Thank you very much for this suggestion. The required correction was made. See line 27.

- Line 14 “High sensory scores”: Indicate numerical values.

Thank you very much for this suggestion. The required data was provided. See lines 26

- Lines 26-27: Please replace with a more accurate conclusion of the work.

Thank you very much for this suggestion. The required correction was made. See lines 32-34.

- Lines 32-34: Explain better the issue related to celiac disease. Check other information at https://doi.org/10.1007/s11947-022-02975-1

Thank you very much for this suggestion. The required data was provided. See lines 42-49.

- Lines 40-44: Indicate the benefits of consumption of these components.

Thank you very much for this suggestion. The required data was provided. See lines 61-64.

- Line 69: Check the term “very low”, since more than 50% of the okara is reused, a much higher amount than other by-products.

Thank you very much for this suggestion. The sentence is rewritten. See lines 89-90.

- Introduction: Include a brief technological explanation regarding the role of ingredients and the challenges of substitution in gluten-free matrices.

Thank you very much for this suggestion. The required data was provided. See lines 113-117; 121-126

- Lines 72-77: Indicate textually what explains such a large variation in the components.

Thank you very much for this suggestion. The required data was provided. See lines 102-106.

- lines 82-83: Please review the sentence, as some products have already been prepared with the addition of okara. Indicate the differential in relation to the current study (check the excellent explanation of lines 84-91):

10.1007/s00217-019-03261-9

10.15406/mojfpt.2017.04.00111 

 Thank you very much for this suggestion. The required data was provided. See lines 121-126

And others

- Line 86: Change “conent” to “content”.

Thank you very much for this suggestion. The required correction was made, line 129.

- 4. Material and methods: Check the correct order of presentation of the methodology. Other steps were performed before the sensory analysis of the bread.

Thank you very much for this suggestion. The required corrections were made.

- Line 462: Please check the correct citation form.

Thank you very much for this observation. The correction was made See line 584.

- Lines 469-470: Check the sentence, as some information about the procedure is missing o.

Thank you very much for this suggestion. The sentence is reworded. See lines 591-592

- Line 473: salt and water (160%). Is this information correct?

Thank you very much for this observation. The missing information was added. See lines 595.

- Lines 113-130: Add possible explanations of matrices or other steps that can justify the opposite of what is verified in the literature for the influence of okara in the matrix of bread.

Thank you very much for this suggestion. The additional information was provided. See line 180-187.

-  Lines 139-144: Describe whether this has been verified in other works.

Thank you very much for this suggestion. The additional information was provided. See line 201-204.

- Line 156: Check the correct writing of the sentence “parameters; Table 2).”

Thank you very much for this suggestion. However, table 2 and corresponding text were removed according to the suggestion of other reviewer.

- Lines 207-208: Specify better what corresponds to the appropriate technological processes.

Thank you very much for this suggestion. The additional information is provided. See line 302-303.

- Line 229: Check the sentence “carbohydrates, Figure 2)”.

Thank you very much for this suggestion. The sentence was reworded. See lines 322-323.

- Lines245-248: Please check the sentence, applied salt concentration is low for microbial inhibition.

Thank you very much for this suggestion. The data were checked. Values of salt concentration 0,2% - 2,5% were cited from the reference Šmídová and Rysová (2022), page 3, table 1.

- Change “FRP” to “FRAP” derived from Ferric Reducing Antioxidant Power Assay”.

In this study FRP-ferric reducing power assay was used. See reference Milinčić et al. (2022).

- Lines 275-277: Better explain the use of different methods, which are based on the characteristics of the various components, such as polarities, and categories, among others.

Thank you very much for this suggestion. The additional information was provided. See line 373-380.

- Antioxidant activity: Please compare with synthetic antioxidants widely reported in the literature.

Thank you very much for this suggestion. The additional information was provided. See line 525-531

- 2.4. Macro- and microelements: improve the discussion based on the role of minerals in the body or technological function (rather than concentration comparison with other work).

Thank you very much for this suggestion. The additional information was provided. See lines 399-429

- Lines 359-364: And how can this be resolved?

Thank you very much for this remark. The calculation error has been made and the values for Na were presented 10 times higher. The correction of DRI values for Na was done, table 8. The last paragraph of 2.5 section was deleted.

Gómez, M. (2022). Chapter Five - Gluten-free bakery products: Ingredients and processes. In W. Zhou & J. Gao (Eds.), Advances in Food and Nutrition Research (Vol. 99, pp. 189-238). Academic Press. https://doi.org/https://doi.org/10.1016/bs.afnr.2021.11.005

Guimarães, R. M., Pimentel, T. C., de Rezende, T. A. M., Silva, J. d. S., Falcão, H. G., Ida, E. I., & Egea, M. B. (2019). Gluten-free bread: effect of soy and corn co-products on the quality parameters. European Food Research and Technology, 245(7), 1365-1376. https://doi.org/10.1007/s00217-019-03261-9

Ioniță-Mîndrican, C. B., Ziani, K., Mititelu, M., Oprea, E., Neacșu, S. M., Moroșan, E., Dumitrescu, D. E., Roșca, A. C., Drăgănescu, D., & Negrei, C. (2022). Therapeutic Benefits and Dietary Restrictions of Fiber Intake: A State of the Art Review. Nutrients, 14(13). https://doi.org/10.3390/nu14132641

Magano, N., du Rand, G., & de Kock, H. (2022). Perception of Gluten-Free Bread as Influenced by Information and Health and Taste Attitudes of Millennials. Foods, 11(4), 491. https://www.mdpi.com/2304-8158/11/4/491 

Milinčić, D. D., Stanisavljević, N. S., Kostić, A. Ž., Gašić, U. M., Stanojević, S. P., Tešić, Ž. L., & Pešić, M. B. (2022). Bioaccessibility of Phenolic Compounds and Antioxidant Properties of Goat-Milk Powder Fortified with Grape-Pomace-Seed Extract after In Vitro Gastrointestinal Digestion. Antioxidants, 11(11), 2164. https://doi.org/doi:10.3390/antiox11112164

Moore, M. M., Schober, T. J., Dockery, P., & Arendt, E. K. (2004). Textural Comparisons of Gluten-Free and Wheat-Based Doughs, Batters, and Breads. Cereal Chemistry, 81(5), 567-575. https://doi.org/https://doi.org/10.1094/CCHEM.2004.81.5.567

Sandri, L. T. B., Santos, F. G., Fratelli, C., & Capriles, V. D. (2017). Development of gluten-free bread formulations containing whole chia flour with acceptable sensory properties. Food Science & Nutrition, 5(5), 1021-1028. https://doi.org/https://doi.org/10.1002/fsn3.495

Šmídová, Z., & Rysová, J. (2022). Gluten-Free Bread and Bakery Products Technology. Foods, 11(3), 480. https://www.mdpi.com/2304-8158/11/3/480 

Stanojevic, S. P., Barac, M. B., Pesic, M. B., Zilic, S. M., Kresovic, M. M., & Vucelic-Radovic, B. V. (2014). Mineral elements, lipoxygenase activity, and antioxidant capacity of okara as a byproduct in hydrothermal processing of soy milk  [Article]. Journal of Agricultural and Food Chemistry, 62(36), 9017-9023. https://doi.org/10.1021/jf501800s

Round 2

Reviewer 1 Report

The authors have addressed my concerns.

Author Response

R1: The authors have addressed my concerns.

Thank you very much for your opinion and all of your work done to improve the quality of the manuscript. We are glad that all issues raised were addressed properly.

Reviewer 2 Report

Icono de Validado por la comunidad

In the table 5, values ​​of the correlation coefficient equal to 1 are obtained. Has only 1 sample been analyzed? That correlation doesn't make sense. What is the reason for the negative correlation between total phenolic compounds and antioxidant capacity measured by the ABTS method?

Why is the composition of all the samples not analyzed and not only the best accepted by the consumers?

Author Response

RESPONSES TO REVIEWER 2 COMMENTS

Thank you very much for your comments. All comments are addressed below.

In the table 5, values ​​of the correlation coefficient equal to 1 are obtained. Has only 1 sample been analyzed? That correlation doesn't make sense. What is the reason for the negative correlation between total phenolic compounds and antioxidant capacity measured by the ABTS method?

Thank you very much for this observation. We agree with you. The sample size should be higher to obtain more convenient results. Table 5 and the corresponding discussion were deleted from the text.    

Why is the composition of all the samples not analyzed and not only the best accepted by the consumers?

The primary goal of this research was to prepare sensory acceptable bread for consumers. This task is very demanding considering that the majority of consumers react negatively to the sensory characteristics of gluten-free bread as well as to the unfavorable leguminous properties of soy-food, that are generally unacceptable for most consumers of the Western market (Guimarães et al., 2019; Sandri et al., 2017). 

So the work plan was to formulate gluten-free bread acceptable for the consumers and then conduct physicochemical and functional property analysis only for the sensory most acceptable bread.

For clarification, an additional sentence in green letters was added in the section Results and Discussion, lines 140-144.

Guimarães, R. M., Pimentel, T. C., de Rezende, T. A. M., Silva, J. d. S., Falcão, H. G., Ida, E. I., & Egea, M. B. (2019). Gluten-free bread: effect of soy and corn co-products on the quality parameters. European Food Research and Technology, 245(7), 1365-1376. https://doi.org/10.1007/s00217-019-03261-9

Sandri, L. T. B., Santos, F. G., Fratelli, C., & Capriles, V. D. (2017). Development of gluten-free bread formulations containing whole chia flour with acceptable sensory properties. Food Science & Nutrition, 5(5), 1021-1028. https://doi.org/https://doi.org/10.1002/fsn3.495